# Laser-Heat Surface Treatment of Superwetting Copper Foam for Efficient Oil–Water Separation

**DOI:** 10.3390/nano13040736

**Published:** 2023-02-15

**Authors:** Qinghua Wang, Chao Liu, Huixin Wang, Kai Yin, Zhongjie Yu, Taiyuan Wang, Mengqi Ye, Xianjun Pei, Xiaochao Liu

**Affiliations:** 1School of Mechanical Engineering, Southeast University, Nanjing 211189, China; 2State Key Laboratory of High Performance Complex Manufacturing, Central South University, Changsha 410083, China; 3Institute of Agricultural Facilities and Equipment, Jiangsu Academy of Agricultural Sciences, Nanjing 210014, China; 4Key Laboratory of Protected Agriculture Engineering in the Middle and Lower Reaches of Yangtze River, Ministry of Agriculture and Rural Affairs, Nanjing 210014, China

**Keywords:** laser surface treatment, micro/nanostructure, surface chemistry, superwetting, oil–water separation

## Abstract

Oil pollution in the ocean has been a great threaten to human health and the ecological environment, which has raised global concern. Therefore, it is of vital importance to develop simple and efficient techniques for oil–water separation. In this work, a facile and low-cost laser-heat surface treatment method was employed to fabricate superwetting copper (Cu) foam. Nanosecond laser surface texturing was first utilized to generate micro/nanostructures on the skeleton of Cu foam, which would exhibit superhydrophilicity/superoleophilicity. Subsequently, a post-process heat treatment would reduce the surface energy, thus altering the surface chemistry and the surface wettability would be converted to superhydrophobicity/superoleophilicity. With the opposite extreme wetting scenarios in terms of water and oil, the laser-heat treated Cu foam can be applied for oil–water separation and showed high separation efficiency and repeatability. This method can provide a simple and convenient avenue for oil–water separation.

## 1. Introduction

Oil pollution in the ocean, including oil spills and oil diffusion, has become more severe in recent years with the increasing demand of maritime oil transportation, and has raised global concerns since it can also cause significant damage to the ecological environment and human health [1,2,3]. Therefore, it is of vital importance to achieve oil–water separation in order to protect the environment and promote economic sustainability. In the past several decades, a number of different methods have been developed and utilized to realize oil–water separation, including adsorption [4], degradation [5], flocculation [6], in situ burning [7] and oil skimmers [8]. Even though some of these methods have been widely used for industrial applications, they still exhibit distinct drawbacks including high production cost, low processing efficiency, complex procedures and the potential generation of secondary pollution [9,10]. As a consequence, the development of an advanced and efficient oil–water separation method that can overcome the above-mentioned limitations is essentially needed.

Inspired by the plants and animals in nature, a superwetting surface that exhibits either superhydrophobicity [11] or superhydrophilicity [12] could provide a new avenue for the design and fabrication of materials used in oil–water separation. As a matter of fact, superwetting surfaces have been widely studied in recent years, and show a variety of impressive applications including self-cleaning [13], anti-icing [14,15], drag reduction [16] and anti-corrosion [17]. Owing to the fact that oil and water exhibit different surface tensions and densities, membranes with proper porous structure and opposite extreme wetting scenarios in terms of water and oil (e.g., superhydrophobic/superoleophilic or superhydrophilic/superoleophobic) can be manufactured and utilized for effective oil–water separation. For instance, Feng et al. [18] prepared a PTFE-coated metal mesh film with both superhydrophobic and superoleophilic properties and used it for the effective separation of oil and water. Kong et al. [19] fabricated paper-based membranes for oil/water separation by combining an atomic layer deposition of aluminum oxide and, subsequently, coupling of silane molecules to filter papers. Zou et al. [20] prepared composite ceramic membranes by employing fly ash particles recycled from electric plant and kaolin materials and utilized the ceramic membranes for the efficient separation of oil–water emulsions. Zhang et al. [21] constructed a superoleophilic/superhydrophobic surface on copper (Cu) foam using a solution-immersion method, and the foam showed a strong capability for oil–water separation. Although these methods have shown notable effectiveness for oil–water separation, the process complexity and lack of environmental friendliness are nonnegligible constraints that limit their wider applications.

In recent decades, pulsed laser processing using either an ultra-fast pulsed laser [22] or a nanosecond pulsed laser [23,24,25,26,27] has emerged as an alternative option for the fabrication of oil–water separation materials owing to its distinct advantages, including non-contact, ease of control, high process flexibility and selectivity [28,29,30,31]. A pulsed laser beam can precisely and efficiently construct micro/nanostructures on porous membranes including metallic mesh and foam. Combined with the subsequent surface chemistry modification, the laser-textured membrane tends to exhibit opposite extreme wettabilities for oil and water, such as superhydrophobicity/superoleophilicity or superhydrophilicity/superoleophobicity, and thus can be utilized for oil–water separation. Compared with an ultra-fast pulsed laser, a nanosecond laser has gained more popularity for the preparation of oil–water separation materials for its low cost and ease of maintenance, and some research progresses have been achieved in this area. Zhang et al. [32] reported a UV nanosecond pulsed laser-based method to fabricate micro-pore arrays on a Cu sheet. The wettability of the textured Cu sheet was changed to superhydrophobicity/oleophilicity after being exposed to air for 14 days and can be used for oil–water separation. Bakhtiari et al. [33] used a nanosecond fiber laser to drill hole arrays on a brass sheet. Similar to that of Zhang et al., the wettability of the fabricated brass sheet was changed from superhydrophilicity to high hydrophobicity after being exposed to ambient air for 12 days, and then the brass sheet can be used as a filter for the separation of organic solvents from water. Chen et al. [34] drilled hole arrays on an aluminum plate using a nanosecond fiber laser and placed the laser-scanned aluminum plate in a vacuum oven at 100 °C for 24 h. The perforated aluminum plate showed superhydrophobicity/superoleophilicity and was used for effective oil–water separation. Xin et al. [35] prepared super-wetting Cu foam with either superhydrophobicity or superhydrophilicity using nanosecond laser ablation followed by silanization or graphene oxide modification. The laser prepared super-wetting Cu foam can be used to selectively separate heavy oil or light oil from water and exhibited very high separation efficiency up to 99%. Although notable research progress has been achieved for the fabrication of thin metallic sheets or foam with high oil–water separation efficiency using a nanosecond pulsed laser, some key issues still remain and should be resolved. On the one hand, the post-process treatment after laser texturing usually requires several tens of hours to several days for wettability transition that can help to achieve oil–water separation [32,33,34], which has significantly decreased the processing efficiency; on the other hand, chemical reagents such as an FAS-17 solution can achieve fast wettability transition [35], while the reagent can be toxic and expensive, and thus will generate potential hazard to humans and the environment and increase the production cost. Therefore, the development of a nanosecond laser-based method for oil–water separation with high processing efficiency, low cost and environmental friendliness is still of particular interest for the surface engineering community.

In this work, a laser-heat surface treatment method was developed to prepare superhydrophobic/superoleophilic Cu foam for efficient oil water separation. A UV nanosecond laser was firstly employed to induce micro/nanostructures on Cu foam, and subsequently the laser-textured Cu foam was subjected to heat treatment in a conventional oven for 2 h and then ultrasonically cleaned. The surface morphology of the Cu foam was investigated by scanning electron microscope (SEM), and the surface chemistry of the Cu foam was analyzed by energy-dispersive X-ray spectroscopy (EDS). The surface wettability of the Cu foam in terms of both water and oil was studied by water contact angle (WCA) and oil contact angle (OCA) measurements. The influences of some key laser processing parameters, including power, scanning speed and line spacing on the surface morphology and surface wettability of the Cu foam subject to laser-heat surface treatment, was systematically investigated. Finally, the oil–water separation performance of the laser-heat treated Cu foam was evaluated by the oil–water separation test.

## 2. Materials and Methods

### 2.1. Materials

Cu foam with a thickness of 0.19 mm and PPI (pores per linear inch) of 110 was chosen as the experimental material and was purchased from Kunshan GuangJiaYuan New Materials Co. Ltd., Kunshan, China. The test samples were first cut into square pieces with a dimension of 12 × 12 × 0.19 mm, and subsequently, all the samples were ultrasonically cleaned with acetone, ethanol and deionized (DI) water in sequence to eliminate contaminants on the surface. The samples were then stored in ambient air for further treatment and analysis. The soybean oil used in the oil–water separation test was obtained from Linyi Shansong Biological Products Co., Ltd., Linyi, China.

### 2.2. Laser-Heat Surface Treatment

The process schematic for the laser-heat surface treatment method developed in this work can be found in Figure 1. A UV nanosecond laser processing system (MQ5T, Mac Laser, Guangzhou, China) was utilized for surface texturing of the Cu foam. The laser processing system is equipped with a 355 nm UV laser source (Seal-355-3/5, JPT Laser, Shenzhen, China) with a pulse width of 12 ns, an attenuator, a beam expander and a scan head (Sino-Galvo RC1001, Sino-Galvo, Zhenjiang, China). The laser beam was emitted by the UV laser source, and its power intensity and beam diameter could be controlled and modulated by the attenuator and beam expander, respectively. Then the laser beam was focused on the surface of the Cu foam through an f-theta lens with a focal length of 160 nm, and its movement trajectory was designed by CAD software (EzCad, Beijing JCZ Technology Co., Ltd., Beijing, China) and controlled by the scan head. A cooling system is connected to the scan head, which can prevent the scan head from overheating during the laser surface texturing. The laser processing parameters utilized and investigated in this work can be found in Table 1, and the range of these parameters was determined to be befitting for the developed method after conducting a set of preliminary experiments.

The laser-textured Cu foam exhibited superhydrophilicity and superoleophilicity. To achieve rapid wettability conversion from superhydrophilicity to superhydrophobicity, a post-process treatment was applied, as shown in Figure 1. The laser-textured Cu was first placed in a conventional oven at 200 °C for 2 h, and then ultrasonically cleaned with ethanol and DI water successively. With the postprocess treatment, the laser-heat treated Cu foam exhibited superhydrophobicity and superoleophilicity, and thus can be utilized for oil–water separation. In this work, four types of Cu foams were prepared for study and analysis: untreated, laser-textured (subject to laser surface texturing only), heat treated (subject to heat treatment only) and laser-heated treated (combining laser surface texturing and heat treatment).

### 2.3. Surface Characterizations

The surface morphology of the untreated Cu foam and the laser-heat treated Cu foams was observed using a field emission scanning electron microscope (FESEM, FEI Sirion, Hillsboro, OR, USA). The surface chemistry of the untreated Cu foam, laser-textured Cu foam and laser-heat treated Cu foam was evaluated using energy-dispersive X-ray spectroscopy (EDX, FEI Sirion, Hillsboro, OR, USA). The surface wettability was examined via a water contact angle (WCA) measurement and an oil contact angle (OCA) measurement using a contact angle goniometer (SCA-100, Mumuxili Technology, Nanjing, China) furnished with a high-resolution CMOS camera. During each WCA or OCA measurement, a water droplet or an oil droplet with a volume of ~4 µL was dropped onto the sample. The optical shadowgraph was captured for each measurement, and image analysis software was utilized to determine the WCA or OCA value. Five WCA or OCA measurements were conducted for each sample, and the average WCA or OCA value was reported.

### 2.4. Oil–Water Separation

A self-designed experimental setup was utilized for the oil–water separation test, as shown in Figure 2a. The laser-heated treated Cu foam was fixed between two transparent tubes whose diameter was 20 mm, and a beaker was placed under the tube to collect the separated liquid. Oil and water with colored ink were mixed directly in a volume ratio of 1:1 to prepare the oil–water mixtures, as shown in Figure 2b. During the oil–water separation test, the oil–water mixture was poured onto the laser-treated Cu foam, and the separation process was recorded by a digital camera. The separation efficiency for the oil–water separation process (η) is calculated using the following equation [37]:(1)$$\eta =\frac{m_{1}}{m_{0}}\times 100\%$$
where $m_{1}$ and $m_{0}$ represent the mass of the liquid that has passed through the Cu foam before and after separation, respectively.

## 3. Results

### 3.1. Surface Morphology Analysis

The surface morphology of the untreated Cu foam and laser-heat treated Cu foams was first examined by visual inspection. Figure 3 shows the digital photos of the untreated Cu foam and the laser-heat treated Cu foams processed using different power levels. It can be found that the untreated Cu foam appears to be golden yellow with distinct brightness. Upon laser surface texturing, the laser-heat treated Cu foams turned dark brown due to the strong ablation effect [35], and the color became darker as the power level was increased. This clearly shows that laser surface texturing can cause dramatic change in the color of the Cu foam.

The surface morphology of the untreated Cu foam and laser-heat treated Cu foams was further investigated by SEM, as shown in Figure 4. Figure 4a shows that the untreated Cu foam mainly consisted of integral and smooth skeletons, whose size was around 60~80 µm. There are many layers of skeletons stacked together forming the 3D porous structure. After laser surface texturing, the surface morphology of the laser-heat treated Cu foams was significantly changed. From the low-magnification SEM micrographs shown in Figure 4b–d, it can be seen that the skeleton of the laser-heat treated foam was roughened owing to the influence of pulsed laser irradiation, while its integrity was well retained. With the high-magnification SEM micrographs, it can be more clearly observed that hierarchical micro/nanostructures containing micro-scale Cu foam skeleton and nano-scale particle were successfully formulated. This is mainly attributed to the melting and recasting of Cu foam during laser ablation [24], which leads to the re-deposition of nanoparticles on top of the Cu foam skeleton. It can also be found that as the power level increased, the hierarchy of the surface structure gradually increased. The range of the power level selected in this work appeared to be proper as the hierarchical micro/nanostructures was generated and the structural integrity could be well-maintained simultaneously. Since multi-scale and rough surface structure is a key factor for determining surface wettability [38,39], it is of vital importance to properly select and control the power level for the fabrication of superwetting Cu foam.

Since scanning speed and line spacing are also key parameters for laser surface texturing, their effects on the surface morphology formation of the laser-heat treated Cu foam was also investigated. Figure 5 shows the SEM micrographs for the laser-heat treated Cu foams processed using different combinations of scanning speed and line spacing when the power level of 12 W was utilized. As shown in Figure 5a–d, when the line spacing of 20 µm was utilized, clear hierarchical micro/nanostructures were formed on the Cu foam skeleton, and the scanning speed affected the shape and density of the micro/nanostructures to some extent. As a lower scanning speed was utilized, the Cu foam skeleton became rougher since the number of incident laser pulses per irradiation point increased with the laser beam travelling in a slow speed. This results in a stronger surface melting and re-deposition of the nanoparticles, thus enhancing the surface roughness [40]. Figure 5e,f shows the surface morphology of the laser-heat treated Cu foams processed using the line spacing of 30 µm. It can be found that when the larger line spacing was utilized, the density of the micro/nanostructures distinctly decreased, and the re-deposited particle size dramatically increased. As a result, the hierarchy of the surface structure almost diminished using the line spacing of 30 µm. This is mainly ascribed to the reduction of the laser ablation effect with the increased spacing between the adjacent laser beams. It should be noted that the quality of the laser-induced micro/nanostructures obtained in this work is fairly comparable to those obtained in [35], demostrating the effectiveness of the developed technique for generating a micro/nanostructure on Cu foam. As mentioned earlier, the surface structure is a key parameter that determines surface wettability. Thus, the laser processing parameters including scanning speed and line spacing must be carefully controlled as well to ensure that the laser-induced surface micro/nanostructures with the appropriate density and particle size can be fabricated.

### 3.2. Surface Wettability Analysis

The surface wettability of the Cu foam was evaluated by contact angle measurements in terms of both water and oil. Figure 6 shows the wettability of the Cu foam after each treatment step. The WCA and OCA of the untreated Cu foam are 96.6 ± 1.4° and 0°, respectively, as shown in Figure 6a,b, indicating that the untreated Cu foam was hydrophobic and superoleophilic. The hydrophobicity of the untreated Cu foam can be potentially attributed to the slight decomposition of copper oxide to hydrophobic cuprous oxide [41] or adsorption of hydrophobic organic matter in ambient air [42]. Directly upon laser surface texturing, the Cu foam turned superhydrophilic and superoleophilic with 0° for WCA and OCA, as shown in Figure 6c,d. The dramatic WCA change is attributed to the strong oxidation effect during the laser surface texturing that renders the Cu foam saturated according to the Wenzel regime [43]. If the Cu foam is only heat treated without laser texturing, the Cu foam will exhibit distinct hydrophobicity with a WCA of 120.4 ± 1.8° and superoleophilicity with an OCA of 0° (Figure 6e,f). The WCA increase of the heat-treated Cu foam in comparison with that of the untreated Cu foam can be ascribed to the adsorption of hydrophobic organic species in the air, which has been accelerated by heat treatment [28,36]. However, as the untreated Cu foam is smooth, without any micro/nanostructures, as demonstrated in Figure 4a, it cannot achieve superhydrophobicity [44]. Eventually, with laser-heat surface treatment that combines laser surface texturing and heat treatment, the Cu foam exhibited a wettability transition to superhydrophobicity/superoleophilicity with a WCA of 96.6 ± 1.4° and OCA of 0° (Figure 6g,h), as the Cu foam skeleton had been endowed with both laser-induced micro/nanostructures and a surface chemistry change led by the heat treatment. Figure 7 shows the digital photos of a dripping water droplet and an oil droplet onto the laser-heat treated Cu foam. It can be clearly found that a spherical water droplet formulated on the Cu foam, while the oil droplet quickly spread and penetrated the Cu foam. The distinct difference of the wetting scenarios for water and oil indicates that the laser-heat surface treatment method can effectively prepare superhydrophobic/superoleophilic Cu foam. The surface wettability measurement results in terms of oil and water subjected laser texturing and heat treatment obtained in this work agrees well with that in [34], demonstrating the effectiveness of the developed method for altering surface wettability.

The effect of the laser processing parameters on the wettability transition of the laser-heat treated Cu foam was further investigated. Since the laser-heat treated Cu foam will exhibit superoleophilicity with an OCA of 0° after any type of treatment, only the WCA measurement results are shown and discussed in this section. From Figure 8a, it can be found that the untreated Cu foam was hydrophobic with a WCA slightly higher than 90°. Using the power levels ranging from 8 W to 12 W during laser-heat surface treatment, the laser-heat treated Cu foam was converted to distinct superhydrophobicity with a WCA above 150°. The WCA value increased slightly as the power level increased, which is mainly due to the hierarchy increase of the surface structure, as demonstrated in Figure 4. Furthermore, Figure 8b shows the WCA measurement results for the laser-heat treated Cu foams processed using different combinations of scanning speed and line spacing when a power level of 12 W was utilized. It was discovered that when a line spacing of 20 µm was utilized, the laser-heat treated Cu foam achieved superhydrophobicity, while the scanning speed did not show much impact on the WCA value. However, when a line spacing of 30 µm was utilized, the laser-heat treated Cu foam lost its superhydrophobicity, with a WCA slightly lower than 150°. The wettability difference can be well explained by the SEM images of the surface morphology, as observed in Figure 5. The density of the microstructure was significantly decreased and the particle size became distinctly larger for the laser-heat treated Cu foams when using larger line spacing. Even though all the laser-heat treated Cu foams underwent the same heat treatment process, the difference in the surface structure leads to the reduction of hydrophobicity. Therefore, the laser processing parameters must be well controlled to ensure the wettability transition to superhydrophobicity/superoleophilicity, which is essential for oil–water separation.

### 3.3. Surface Chemistry Analysis

For metallic materials, researchers pointed out that the surface chemistry of the metallic materials can be modified by laser-induced material vaporization and oxidation [45], or heat treatment-led deposition of more carbon-based hydrophobic functional groups [46], thus leading to a surface wettability transition. Therefore, to clearly explore and understand the evolution of surface chemical compositions on the Cu foam before and after laser surface texturing, EDS measurement and analysis was performed as shown in Figure 9, and the detailed atomic elemental compositions can be found in Table 2. The laser-textured Cu foam and laser-heat treated Cu foam considered for surface chemistry analysis in this work were prepared using a power of 12.0 W, a scanning speed of 200 mm/s and a line spacing of 20 µm. As shown in Figure 9a, the untreated Cu foam showed peaks of Cu, C and O, and their atomic percentages were 70.62%, 27.45% and 1.93%, respectively (Table 2). After laser surface texturing, the O peak exhibited a sharp increase as compared to that on the untreated Cu foam, and its atomic percentage increased from 1.93% to 14.80% (Figure 9b and Table 2). This indicates that the Cu foam had been strongly oxidized during the laser surface texturing [35]. As indicated in [47], the metallic oxide can contribute to the formation of hydrogen bonds and increase the surface energy, and the laser-textured Cu foam will thus exhibit superhydrophilicity. Meanwhile, the intensity reduction of the Cu peak on the laser-textured Cu foam could be ascribed to material vaporization. For the laser-heat treated Cu foam, two notable chemical composition changes were discovered, as shown in Figure 9c: on the one hand, the atomic percentage of the C element increased from 29.38% to 32.65% (Table 2). This indicates that more carbon-based organic groups in the air, including –CH_2_–, –CH_3_ and C=C, should have been absorbed onto the laser-heat treated Cu foam [24]. Since these organic groups are hydrophobic, the hydrophobicity of the laser-heat treated Cu foam can be greatly enhanced. On the other hand, a new silicon (Si) peak was clearly detected on the laser-heat treated foam with an atomic percentage of 1.25% (Figure 9c and Table 2). This can represent the formation and deposition of a thin Si-based polydimethylsiloxane (PDMS) layer onto the laser-heat treated Cu foam [36]. The appearance of the thin Si-based PDMS layer can be attributed to the following reason: in this work, a conventional oven was utilized for heat treatment. There is a silicone seal on the door of the oven. During the heat treatment using a temperature of 200 °C, the silicone seal was slightly vaporized, and silicon atoms were deposited onto the laser-treated Cu foam, forming a thin PDMS layer. As the PDMS layer is also known to be hydrophobic [48], the surface wettability of the laser-heat treated Cu foam was further enhanced. These findings agree well with the results of other metallic alloys treated using a similar method [28], and thus it can be believed that the adsorption of more hydrophobic carbon-based organic groups and the deposition of a hydrophobic PDMS layer leads to the realization of superhydrophobicity/superoleophilicity on the laser-heat treated Cu foam.

The EDS elemental mapping data was also obtained to reveal the elemental distributions of the detected elements on each Cu foam. Figure 10a,e,i show the SEM images for the untreated Cu foam, laser-textured Cu foam and laser-heat treated Cu foam where the EDS mapping data was captured. For the untreated Cu foam, it was observed that all the core elements, including Cu, C and O, were uniformly distributed on the skeleton of Cu foam, as indicated in Figure 10b–d. Upon laser surface texturing, a greater color intensity can be observed for the elemental mapping data of O (Figure 10h) as compared with that of the untreated Cu foam, which helps to confirm the strong oxidation during the laser surface texturing. After laser-heat surface treatment, not only did the carbon content absorbed in air increase (Figure 10k), but the newly deposited Si content can be clearly detected (Figure 10m), which corresponds well with the EDS measurement results.

### 3.4. Oil–Water Separation Performance

The oil–water separation performance of the laser-heat treated Cu foam was evaluated by an oil–water separation test. Figure 11 shows the continuous oil–water separation process for the laser-heat treated Cu foam processed using the laser processing parameters of power 12.0 W, scanning speed of 200 mm/s and line spacing of 20 μm. As shown in Figure 11, before pouring the oil–water mixture into the transparent tube, the oil was in the upper part of the beaker, while the water added with color ink was in the lower part. After the pouring process started, the oil was poured into the tube first. Since the laser-heat treated Cu foam exhibits superoleophilicity, the oil easily passed through the Cu foam and dropped into the bottom beaker. When the water started to enter the tube and came in contact with the laser-heat treated Cu foam, it was blocked well by the Cu foam due to its superhydrophobicity. Finally, almost all the oil entered the bottom beaker, while all the water remained blocked in the tube, as clearly shown in Figure 11.

The separation efficiency and repeatability of the laser-heat treated Cu foam were further investigated and analyzed. The separation efficiency of the laser-heat treated Cu foams processed with different laser processing parameters can be found in Figure 12a. It can be found that using a line spacing of 20 µm, the laser-heat treated Cu foams exhibit very good separation efficiency up to 94.6%. The experimental results also indicate that the separation efficiency can improve slightly if the difference between the WCA value and OCA value is larger, indicating that a higher WCA can help to enhance the oil–water separation efficiency. For the laser-heat treated Cu foams processed using the line spacing of 30 µm, the separation efficiency was slightly decreased due to the reduced WCA, while the efficiency can still be higher than 88%. Figure 12b shows the repeatability of separation for the laser-heat treated Cu foam. It was found that as the number of cycles increased, the separation efficiency decreased. However, the efficiency was still higher than 91% after 5 cycles and 87% after 10 cycles. This means that the laser-heat treated Cu foam has good repeatability for oil–water separation, and thus can be recycled even after being used for multiple cycles. Moreover, the separation efficiency and repeatability of oil–water separation for the laser-heat treated Cu foam are fairly comparable to those obtained in [33] and [35]. This further indicates that the laser-heat treated Cu foam fabricated in this work can be well utilized for oil–water separation.

## 4. Conclusions

In this work, a simple, efficient and convenient laser-heat surface treatment method was developed to fabricate superwetting Cu foam that can be used for oil–water separation. The surface morphology, surface wettability surface chemistry and oil–water separation performance of the laser-heat treated Cu foams were studied and analyzed. Some main findings can be summarized as below:Nanosecond laser surface texturing can significantly modify the skeleton structure of the Cu foam by inducing micro/nanostructures on top. The laser processing parameters will also greatly influence the density and size of the micro/nanostructures, and thus should be carefully controlled.The surface energy of the laser-textured Cu foam was reduced by heat treatment, which should be attributed to the combined effects of adsorption of hydrophobic airborne carbon-containing groups and the generation of a Si-based PDMS layer.The Cu foam exhibited superhydrophilicity/superoleophilicity directly upon laser surface texturing and was converted to superhydrophobicity/superoleophilicity subsequent to heat treatment. The wettability transition was ascribed to laser-induced micro/nanostructures and the reduction of surface energy.The laser-heat treated Cu foam showed good performance for oil–water separation with good separation efficiency up to 94.6%, and the oil–water separation process can be repeated.

Overall, the developed method is quite facile and convenient, can potentially be utilized for diverse applications where oil–water separation is needed.

## Figures and Tables

**Figure 1 nanomaterials-13-00736-f001:**
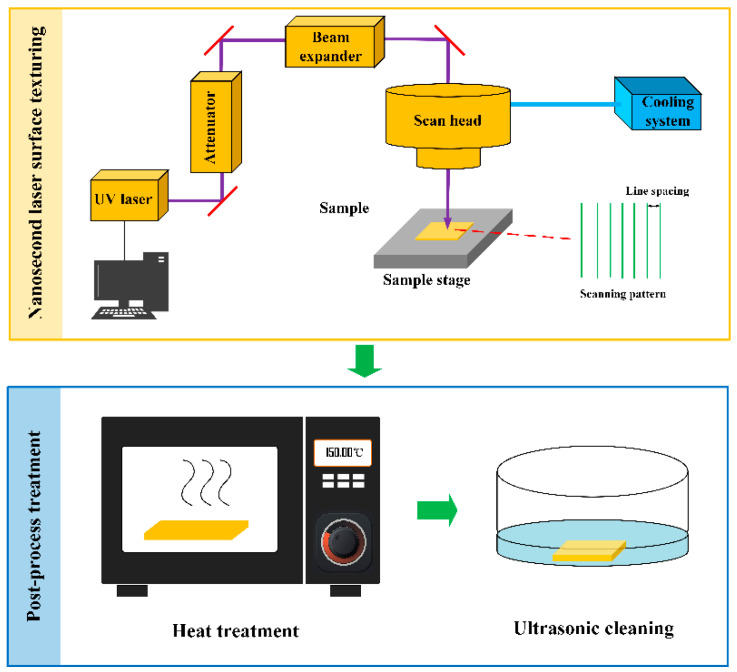
Process schematic for laser-heat surface treatment. Reprinted with permission from ref. [36].

**Figure 2 nanomaterials-13-00736-f002:**
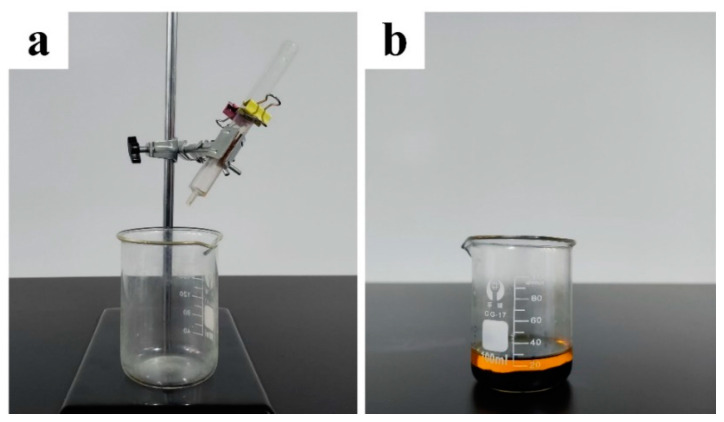
(**a**) Experimental setup for the oil–water separation test; (**b**) the oil–water mixture prepared in this work.

**Figure 3 nanomaterials-13-00736-f003:**
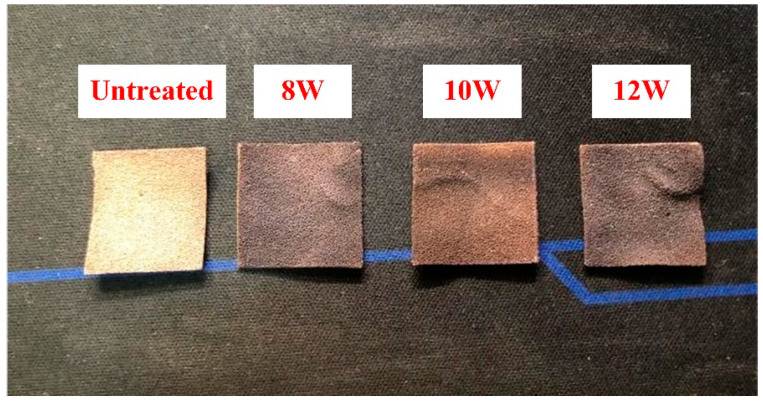
Images of untreated Cu foam and laser-heat treated Cu foams processed with different power levels.

**Figure 4 nanomaterials-13-00736-f004:**
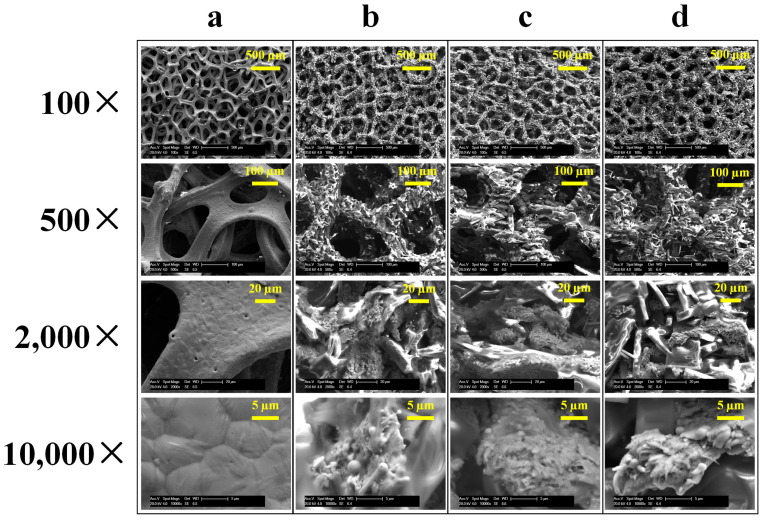
SEM micrographs at different magnifications for (**a**) untreated Cu foam and laser-heat treated Cu foams processed with different power levels (**b**) 8 W; (**c**) 10 W; (**d**) 12 W.

**Figure 5 nanomaterials-13-00736-f005:**
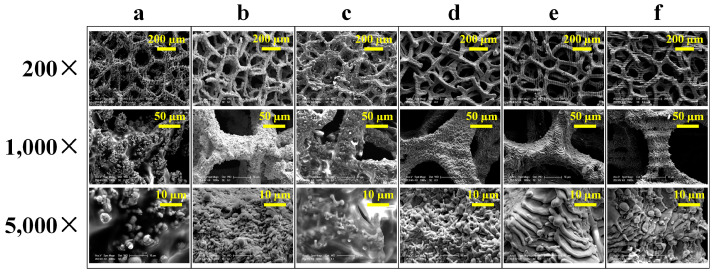
SEM micrographs at different magnifications for laser-heat treated Cu foams processed using different combinations of scanning speed and line spacing: (**a**) scanning speed of 100 mm/s and line spacing of 20 µm; (**b**) scanning speed of 150 mm/s and line spacing of 20 µm; (**c**) scanning speed of 200 mm/s and line spacing of 20 µm; (**d**) scanning speed of 300 mm/s and line spacing of 20 µm; (**e**) scanning speed of 100 mm/s and line spacing of 30 µm; (**f**) scanning speed of 200 mm/s and line spacing of 30 µm.

**Figure 6 nanomaterials-13-00736-f006:**
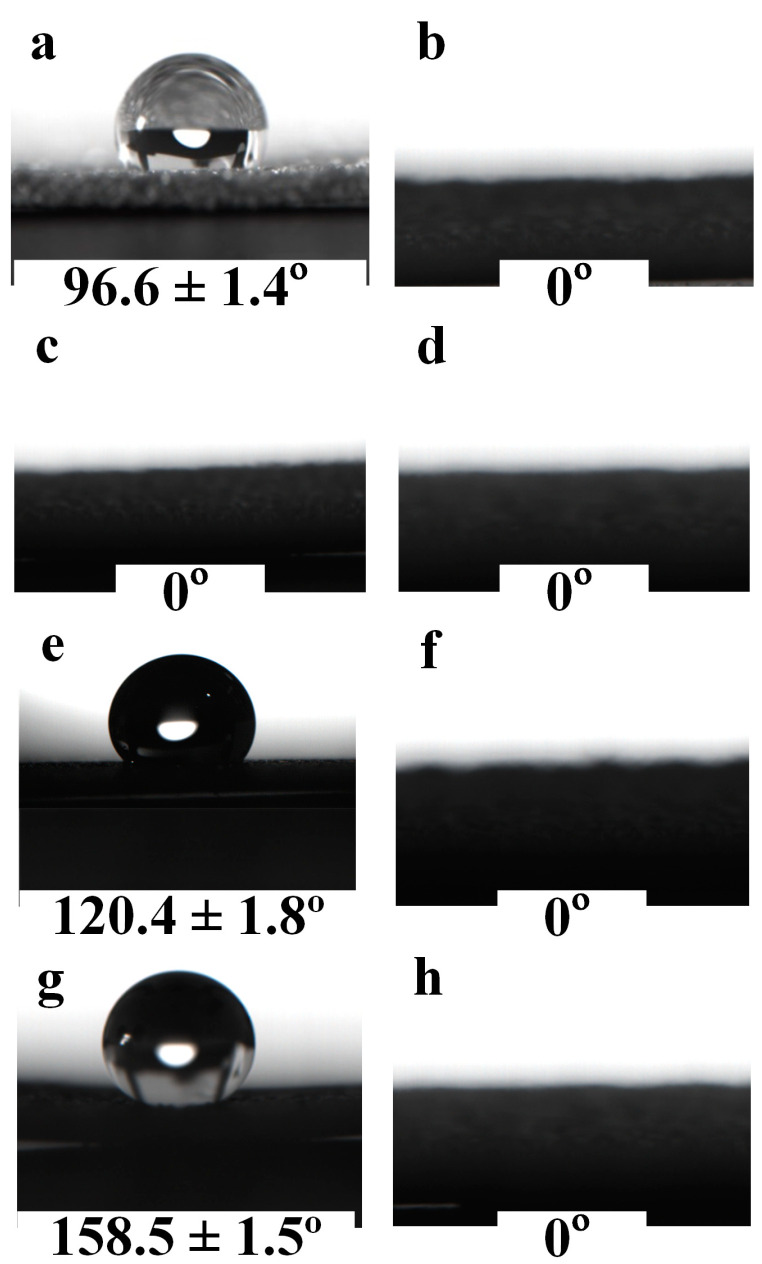
WCA and OCA shadowgraphs for (**a**,**b**) untreated Cu foam; (**c**,**d**) laser-textured Cu foam; (**e**,**f**) heat treated Cu foam and (**g**,**h**) laser-heat treated Cu foam.

**Figure 7 nanomaterials-13-00736-f007:**
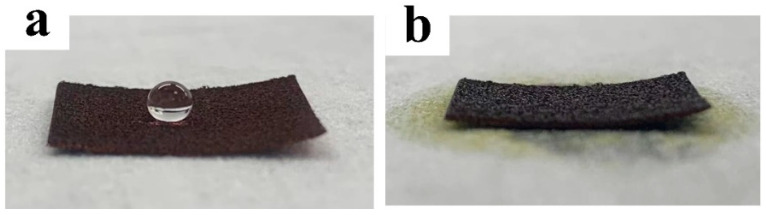
(**a**) Water droplet and (**b**) oil droplet on the laser-heat treated Cu foam indicating its superhydrophobicity/superoleophilicity.

**Figure 8 nanomaterials-13-00736-f008:**
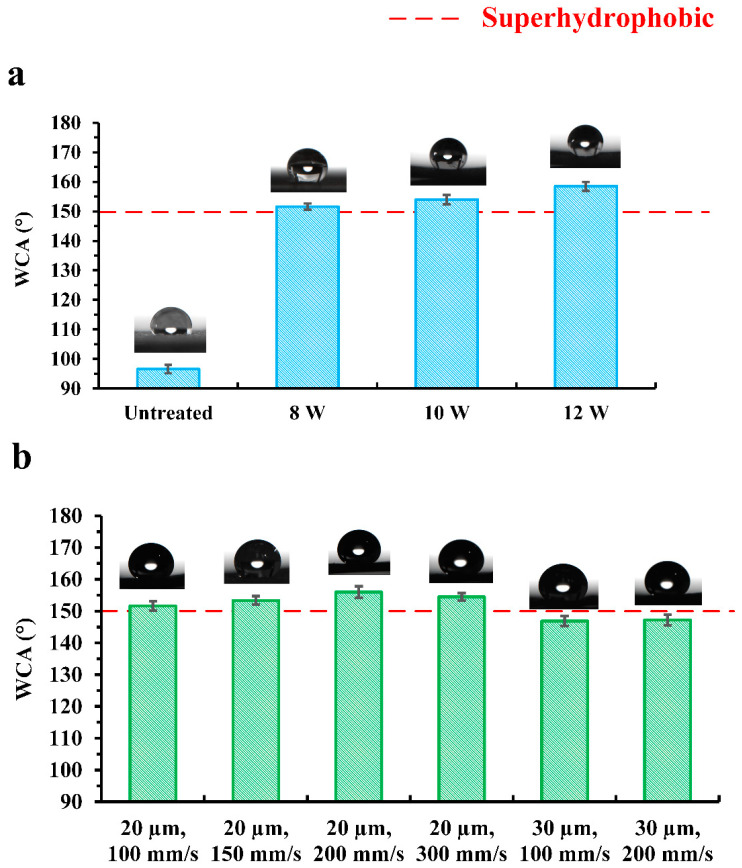
WCA measurement results for (**a**) untreated Cu foam and laser-heat treated Cu foams processed with different power levels and (**b**) laser-heat treated Cu foams processed using different combinations of scanning speed and line spacing.

**Figure 9 nanomaterials-13-00736-f009:**
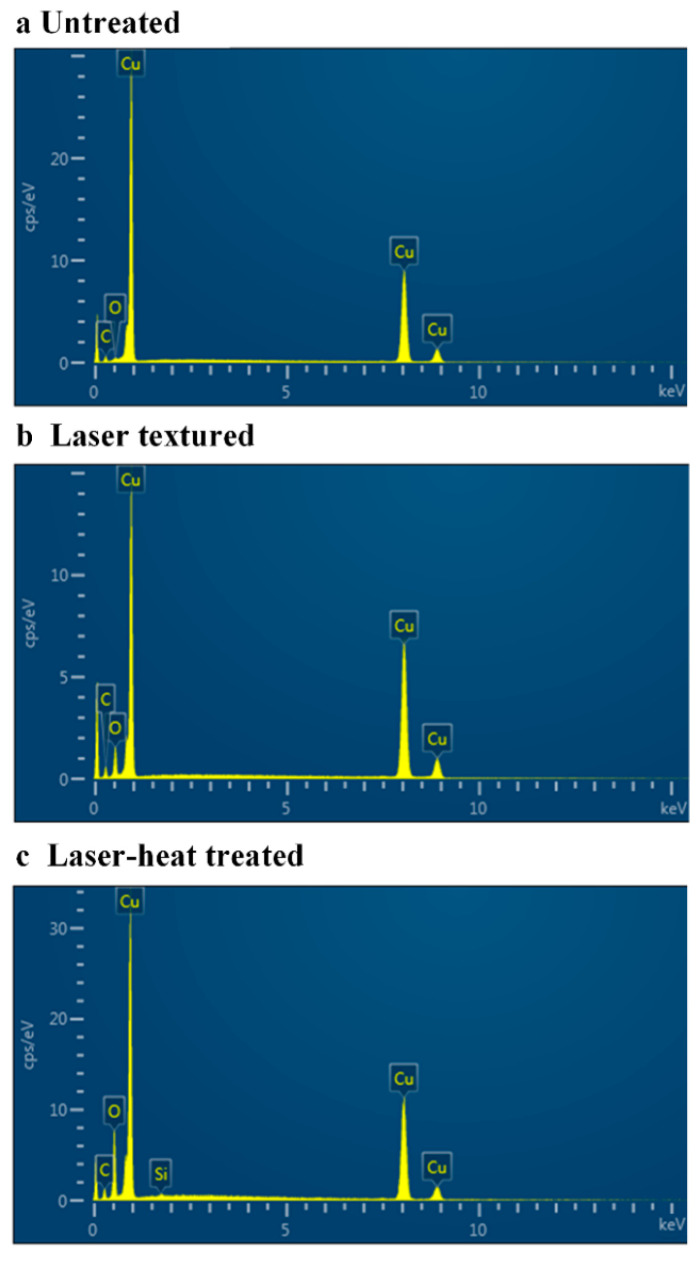
EDS measurement results for (**a**) untreated Cu foam; (**b**) laser-textured Cu foam; (**c**) laser-heat treated Cu foam.

**Figure 10 nanomaterials-13-00736-f010:**
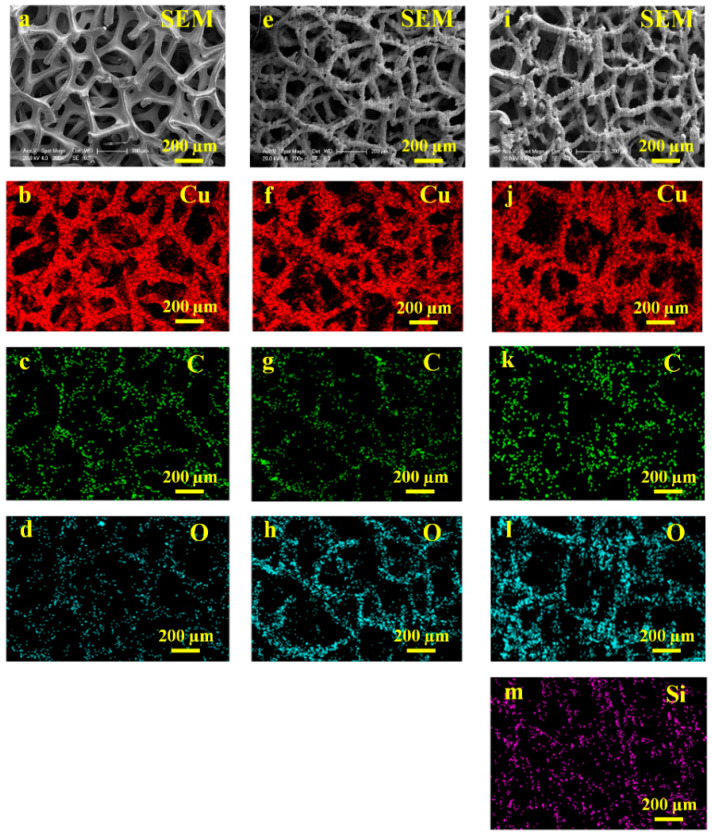
SEM/EDS elemental mapping for (**a**–**d**) untreated Cu foam; (**e**–**h**) laser-textured Cu foam; (**i**–**m**) laser-heat treated Cu foam. (**a**,**e**,**i**) are the SEM images for the corresponding analyzed regions; (**b**–**d**,**f**–**h**,**j**–**m**) are the EDS mapping data showing the qualitative elemental distributions of Cu, C, O and Si.

**Figure 11 nanomaterials-13-00736-f011:**
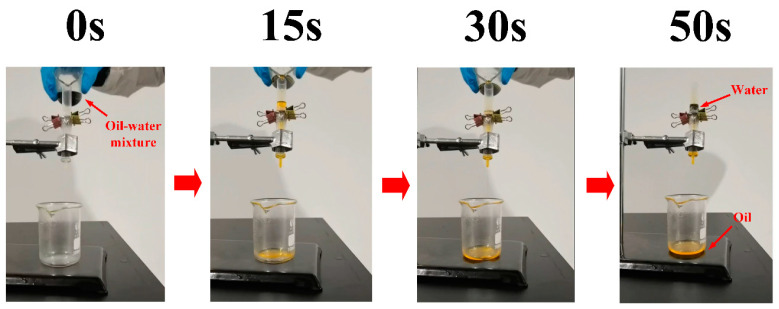
Continuous water/oil separation process using the laser-heat treated Cu foam with the laser processing parameters of power 12.0 W, scanning speed of 200 mm/s and line spacing of 20 μm.

**Figure 12 nanomaterials-13-00736-f012:**
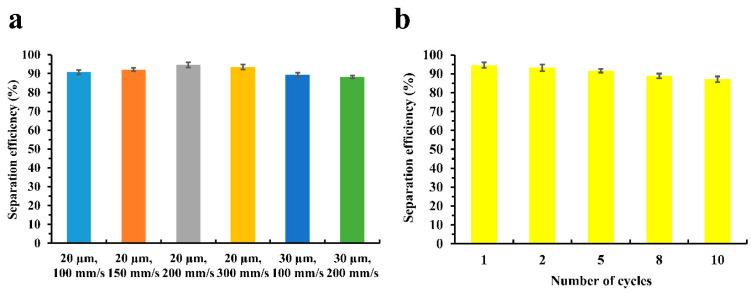
Separation efficiency of the laser-heat treated Cu foams with (**a**) different laser processing parameters and (**b**) multiple cycles.

**Table 1 nanomaterials-13-00736-t001:** Laser processing parameters for laser-heat surface treatment.

Parameters	Value
Average power (W)	8~12
Laser beam diameter (μm)	50
Repetition rate (kHz)	40
Pulse width (ns)	12
Scanning speed (mm/s)	100~300
Line spacing (µm)	20~30
Power intensity (GW/cm^2^)	0.85~1.27
Pulse energy (mJ)	0.20~0.30

**Table 2 nanomaterials-13-00736-t002:** Detailed atomic elemental compositions for untreated Cu foam, laser-textured Cu foam and laser-heat treated Cu foam.

Element (Atomic%)	Untreated	Laser-Textured	Laser-Heat Treated
Cu	70.62	55.82	52.47
C	27.45	29.38	32.65
O	1.93	14.80	13.63
Si	0	0	1.25

## Data Availability

The data presented in this study are available on request from the corresponding author.

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
