# Peer review of "Laser-Heat Surface Treatment of Superwetting Copper Foam for Efficient Oil–Water Separation"

_nanomaterials, 2023, doi:10.3390/nano13040736_

Round 1

Reviewer 1 Report

I have only one request for improvement - in my first comment:

Lines 330-335:   One time you wrote PMDS and two times PDMS. Which abbreviation is the right one? And also, here the abbreviation (PDMS or PMDS) appears in the paper for the first time, so please write it with full words also, at the place where it is used for the first time. And this requires a minor correction of your paper: please explain in the discussion where the Si came from?

Lines 303-336: This is just a comment: during surface texturing Cu foam is oxidised and would thus become more hydrophilic. But, increase of hydrophobicity has been attributed to the formation of various deposits that are hydrophobic and these are prevailing. Have I understood this right?

Author Response

Comment: 

I have only one request for improvement - in my first comment:

Lines 330-335:   One time you wrote PMDS and two times PDMS. Which abbreviation is the right one? And also, here the abbreviation (PDMS or PMDS) appears in the paper for the first time, so please write it with full words also, at the place where it is used for the first time. And this requires a minor correction of your paper: please explain in the discussion where the Si came from?

Response:

Thanks for the reviewer’s valuable comments. Below is our response to each point: (1) The correct terminology should be “PDMS”. “PMDS” is a typo. Thanks for pointing out the error. The typos have been corrected in the manuscript; (2) The full name of “PDMS” is “polydimethylsiloxane” and it has been added at the place where it is used for the first time; (3) In this work, a conventional oven was utilized for heat treatment. There is a silicone seal on the door of the oven. During the heat treatment using a temperature of 200 ºC, the silicone seal was slightly vaporized and silicon atoms were deposited onto the laser treated Cu foam forming a thin PDMS layer. This is how the Si element was originated. Since the silicon-based organic polymer PDMS is known to be hydrophobic, it could further enhance the hydrophobicity of the laser textured Cu foam and render superhydrophobicity. The explanation has been added to the revised manuscript.

Comment:

Lines 303-336: This is just a comment: during surface texturing Cu foam is oxidised and would thus become more hydrophilic. But, increase of hydrophobicity has been attributed to the formation of various deposits that are hydrophobic and these are prevailing. Have I understood this right?

Response:

Thanks for the reviewer’s valuable comments. The understanding is correct. Two factors are essentially needed for superhydrophobicity: surface micro/nanostructures and proper surface chemistry. Laser surface texturing would generate surface micro/nanostructures on the Cu foam, while oxidize the surface and increase surface energy leading to superhydrophilicity. Later, heat treatment was utilized to change the surface chemistry by depositing hydrophobic carbon-based organic groups and a thin silicon-based PDMS layer onto the surface. This surface chemistry change can significantly reduce the surface energy and result in wettability conversion from superhydrophilicity to superhydrophobicity.

Reviewer 2 Report

The Authors described studies concerning on laser heating of surface of superwetting copper foam for oil-water separation. The aim of present manuscript is shown quite clear in introduction. Moreover, I the experimental section is written quite good. I have only one doubt Please improve the Discussion. Please try to compare your results to other researchers

Generally, the Authors did some work. However it could not be published in present form in Nanomaterials. I recommend minor revision.

Author Response

Comment:

The Authors described studies concerning on laser heating of surface of superwetting copper foam for oil-water separation. The aim of present manuscript is shown quite clear in introduction. Moreover, I the experimental section is written quite good. I have only one doubt Please improve the Discussion. Please try to compare your results to other researchers

Generally, the Authors did some work. However, it could not be published in present form in Nanomaterials. I recommend minor revision.

Reponse:

Thanks for the reviewer’s valuable comments. We have added the comparison of the experimental results obtained by this work and by other researchers in the Discussion section. The comparison has been conducted in terms of surface structure, surface wettability, surface chemistry and oil-water separation.

Reviewer 3 Report

The article "Laser-heat surface treatment of superwetting copper foam for efficient oil-water separation" deals with the study of the laser-heat treated Cu foam can be applied for oil-water separation. The authors showed high separation efficiency and repeatability. Descibed method can provide a simple and convenient avenue for oil-water separation. The title of the article corresponds to its content, and the subject matter is consistent with the profile of the magazine. The authors indicate the purpose of the research and its results. References are well selected, they reflect the works that has been carried out in the world in recent years.

It is not clear to me, from where the Si atoms appeared in the Cu foam after laser treatment. There is no explanation for this issue, PDMS is mentioned 2 times as a hydrophobic material, but the authors do not give any source of siloxane or Si. This is what is most missing in this work and needs to be supplemented.

This manuscript seems to be useful to the readers working on multifunctional superhydrophobic/superoleophobic materials. I think that after supplementing  , the manuscript is suitable for publication in Nanomaterials.

Author Response

Comment:

The article "Laser-heat surface treatment of superwetting copper foam for efficient oil-water separation" deals with the study of the laser-heat treated Cu foam can be applied for oil-water separation. The authors showed high separation efficiency and repeatability. Descibed method can provide a simple and convenient avenue for oil-water separation. The title of the article corresponds to its content, and the subject matter is consistent with the profile of the magazine. The authors indicate the purpose of the research and its results. References are well selected, they reflect the works that has been carried out in the world in recent years.

It is not clear to me, from where the Si atoms appeared in the Cu foam after laser treatment. There is no explanation for this issue, PDMS is mentioned 2 times as a hydrophobic material, but the authors do not give any source of siloxane or Si. This is what is most missing in this work and needs to be supplemented.

Response:

Thanks for the reviewer’s valuable comments. Below is the explanation for the source of Si elements on the laser textured Cu foam after heat treatment: In this work, a conventional oven was utilized for heat treatment. There is a silicone seal on the door of the oven. During the heat treatment using a temperature of 200 ºC, the silicone seal was slightly vaporized and silicon atoms were deposited onto the laser treated Cu foam forming a thin PDMS layer. This is how the Si element was originated. Since the silicon-based organic polymer PDMS is known to be hydrophobic, it could further enhance the hydrophobicity of the laser textured Cu foam and render superhydrophobicity. The explanation has been added to the revised manuscript.

Comment:

This manuscript seems to be useful to the readers working on multifunctional superhydrophobic/superoleophobic materials. I think that after supplementing, the manuscript is suitable for publication in Nanomaterials.

Reply:

Thanks for the reviewer’s valuable comments. We have provided the supplementing explanation as required by the reviewer. We hope that the explanation has been clear enough that can satisfy the requirement of the reviewer.
